# Indicators for Measuring Intergenerational Fairness of Social Security Systems—The Case of the German Social Health Insurance

Stefan Fetzer [1,*] and Stefan Moog [2]

1. Faculty of Business, Aalen University of Applied Sciences, D-73430 Aalen, Germany
2. Prognos AG, D-79100 Freiburg im Breisgau, Germany; Stefan.Moog@prognos.com
* Correspondence: Stefan.Fetzer@hs-aalen.de

**Abstract:** The issue of fiscal sustainability is often labelled as a synonym for intergenerational fairness; however, pay-as-you-go schemes such as the German Social Health Insurance (SHI) involve a "natural" amount of intergenerational redistribution from younger net payers to older net beneficiaries. We calculate intertemporal balance sheets of SHI and compare two generational accounting approaches (GAC and GAIB) with an alternative measure of intergenerational fairness, SM, which we derive from Settergren and Mikula (2005). Our results indicate that the SM concept leads to similar implications concerning the amount of intergenerational redistribution as classical measures of fiscal sustainability. For the SM approach, the balance sheet of SHI shows a rate of unfunded benefits of 25 percent. Closing this gap requires an increase of the contribution rate by 30 to 40 percent. This total effect can be separated into an effect due to the current population structure (10 p.p.), the increase in life expectancy (10 p.p.), and medical technical progress (about 10 to 20 p.p.).

**Keywords:** social health insurance; fiscal sustainability; intergenerational fairness; generational accounting; turnover duration; demographic transition; medical progress

## 1. Introduction

In the early 1990s, economists developed quantitative methods to empirically assess the long-term implications of demographic transition on government and social security budgets. So far, the most prominent approaches have been the method of Generational Accounting developed by Auerbach, Gokhale, and Kotlikoff [1–3] and the OECD method based on the concept of fiscal sustainability proposed by Blanchard et al. [4]. The issue of (fiscal) sustainability is often labelled as a synonym for intergenerational fairness, i.e., a sustainable fiscal policy implies quasi-fairness between current and future generations [5].

However, in the case of pay-as-you-go schemes, such statements about intergenerational fairness should be interpreted with caution. A key characteristic of pay-as-you-go schemes is that they involve a "natural" amount of intergenerational redistribution from younger net payers to older net beneficiaries. Previously conducted studies usually do not take this "natural" amount of intergenerational redistribution explicitly into account.

In this paper, we consider an alternative measure of intergenerational fairness, derived from Settergren and Mikula [6], which provides the theoretical foundation of the automatic balance mechanism in the Swedish pension scheme [7] and compares it with two generational accounting approaches. Most applications of the Settergren and Mikula [6] approach are concerned with the fiscal sustainability of public pension systems (e.g., [8–11]). So far, it has rarely been applied to other social security systems.

For our case study, we use the German Statutory Health Insurance (SHI), a public health insurance scheme in the Bismarckian tradition. Similar to pay-as-you-go pension schemes, the financing of German SHI relies on contributions mainly made by workers, whereas benefits are received by pensioners for their most part. As Germany is one of

the fastest aging societies in the world, the demographic transition will lead to a growing financial gap between the average contribution payments and average benefits.

Furthermore, even without the increasing burden of the demographic transition, it is to be expected that health care expenditures will outgrow GDP per capita due to medical technical progress. In the last decades the health expenditures of OECD countries, but also those of low and middle income states such as the BRICS states, achieved excess growth rates of 0.5 to 1.5 percentage points per year [12–16]. Hence, in our analysis we also consider a growth difference between contributions and benefits. Thereby we expand the (traditional) theoretical framework of Settergren and Mikula [6].

For the comparison of our measure on basis of Settergren and Mikula [6] with the two generational accounting approaches, we prepare intertemporal balance sheets and derive comparable indicators to measure the amount of intergenerational distribution in SHI due to the demographic transition and medical progress. Furthermore, we test the sensitivity of these indicators to differing assumptions regarding the demographic development as well as the chosen parameters for growth and interest.

This paper is organized as follows: Section 2 presents the theoretical background of our measuring concepts. Section 3 shows the results for the German Statutory Health Insurance scheme (SHI). Section 4 summarizes and discusses our findings, and Section 5 draws a conclusion.

## 2. Methods and Data

### 2.1. Intertemporal Balance Sheets

At first glance, the financial design of the German SHI is similar to a defined benefit pension system. However, in contrast to a pension system with a defined benefit formula, the benefit-in-kind principle applies to the majority of reimbursed services of SHI [17]. As a consequence, the future development of SHI benefits depends on many factors and is more uncertain. These factors include the future generosity of the SHI system itself (Which medical services will be reimbursed in what amount?), the future demand for health services (How will the spectrum of diseases evolve? Will there be an increasing preference for health services on an individual level?), and the future supply of medical goods (Will there be expensive new medical treatments?).

Due to the lack of a "benefit formula for SHI benefits", we choose a pragmatic approach for this case study and distinguish two scenarios concerning the future development of benefits. In the first scenario, we assume that benefit receipts grow by the same annual rate as contribution payments (status quo principle). The second scenario mimics the medical technical progress by assuming more sharply rising benefits.

Our starting point is an intertemporal balance sheet of SHI (Table 1). It shows similarities and differences between our approach based on Settergren and Mikula [6], *SM*, and two generational accounting approaches, labelled as Generational Accounting (classical), *GAC*, and Generational Accounting (implicit burden concept), *GAIB*. For each measure, the initial intertemporal balance sheet displays the assets of SHI on the left side and its liabilities on the right side. An intertemporal deficit occurs when the liabilities exceed the assets. As our indicator for intergenerational fairness, we consider the increase in contribution payments necessary to close (or balance) the intertemporal balance sheet (for the construction of this indicator in the context of fiscal sustainability measurement, see [18,19]). Put differently, by intergenerational fairness we mean a closed balanced sheet as finally shown in Table 1.

**Table 1.** Intertemporal balance sheets of different measuring concepts.

| SM Concept | | GAC Concept | | GAIB Concept | |
|---|---|---|---|---|---|
| **Initial intertemporal balance sheet** | | | | | |
| Assets | Liabilities | Assets | Liabilities | Assets | Liabilities |
| *PVCL* | *PVBL* | *PVCL* | *PVBL* | *PVCL* | *PVBL* |
| *CA* | | *PVCF* | *PVBF* | *PVadjCL* | |
| *UF* | | *IPL* | | *IB* | |
| **Indicator for intergenerational fairness** | | | | | |
| $\alpha = \frac{UF}{PVCL+CA}$ | | $\theta = \frac{IPL}{PVCL+PVCF}$ | | $\beta = \frac{IB}{PVCL+PVadjCL}$ | |
| **Closed intertemporal balance sheet** | | | | | |
| Assets | Liabilities | Assets | Liabilities | Assets | Liabilities |
| $(1+\alpha)$ $\cdot PVCL$ | *PVBL* | $(1+\theta)$ $\cdot PVCL$ | *PVBL* | $(1+\beta)$ $\cdot PVCL$ | *PVBL* |
| $(1+\alpha)\cdot CA$ | | $(1+\theta)$ $\cdot PVCF$ | *PVBF* | $(1+\beta)\cdot PVadjCL$ | |

For each concept, the intertemporal balance sheet includes the present value of future contribution payments, *PVCL*, and future benefits, *PVBL*, of all cohorts alive in the base year *BY*. They can be expressed as follows:

$$PVCL = \sum_{x=0}^{100} N_{x,BY} \sum_{u=x}^{100} ndf^{u-x} \cdot l_{x,BY,u} \cdot c_u \tag{1}$$

$$PVBL = \sum_{x=0}^{100} N_{x,BY} \sum_{u=x}^{100} ndf^{u-x} \cdot l_{x,BY,u} \cdot (1+cp)^{u-x} \cdot b_u \tag{2}$$

Here, $N_{x,BY}$ denotes the total number of insured persons at age $x$ in the base year *BY*, and $ndf = (1+g)/(1+r)$ is the net discounting factor given by the ratio of the annual rate of productivity growth $g$ and the annual interest rate $r$. $l_{x,BY,u}$ represents the conditional survival probability that a person at age $x$ in the base year *BY* will be alive at age $u$. $c_u$ and $b_u$ stand for the contribution payment and the benefit receipt at age $u$, respectively. Note that both $c_u$ and $b_u$ are assumed to be constant over time, which means that (in addition to the effect of productivity growth accounted for in $ndf$) the average contribution payment or benefit receipt of a person at age $u$ in the future is the same as for a person at age $u$ in the base year *BY*. Finally, as we expand our model to take into account the consequences of medical progress, we consider the possibility of a growth differential between contribution payments and benefit receipts, which is denoted by $cp$.

Except for the two items *PVCL* and *PVBL*, Table 1 shows that the initial balance sheets differ for each of the three measurement concepts. In case of the *SM* concept, the balance sheet includes one additional item of SHI assets, a contribution asset, *CA*, which can be interpreted as the permanent flow of contribution payments by future cohorts inherent in the SHI scheme or more generally any pay-as-you-go scheme. Put differently, *CA* can be interpreted as a measure of the "natural" amount of intergenerational redistribution of a pay-as-you-go scheme. As shown by Settergren and Mikula [6], the contribution asset can be calculated as the total amount of base year contribution payments, $Contributions_{BY}$, times the so-called turnover duration, *TD*:

$$CA = TD \cdot Contributions_{BY} \tag{3}$$

Under the assumption of no population growth, a discount rate that equals the growth rate of contributions and benefits the turnover duration reflects the difference of the average value-weighted average age of beneficiaries, $A^B$, and contributors, $A^C$, in the base year:

$$TD = A^B - A^C \tag{4}$$

$$\text{whereby } A^B = \frac{\sum_{x=0}^{100} x \cdot N_x \cdot b_x}{\sum_{x=0}^{100} N_x \cdot b_x} \tag{5}$$

$$\text{and } A^C = \frac{\sum_{x=0}^{100} x \cdot N_x \cdot C_x}{\sum_{x=0}^{100} N_x \cdot C_x} \tag{6}$$

The intertemporal deficit or underfunding $UF$ in the $SM$ concept is therefore given by:

$$UF = PVBL - PVCL - CA \tag{7}$$

Our indicator for intergenerational fairness, the increase of contributions necessary to balance intertemporal assets and liabilities, $\alpha$, can be calculated as:

$$\alpha = \frac{UF}{PVCL + CA} \tag{8}$$

With the closed balance sheet, we reach the condition for intergenerational fairness, described in Settergren and Mikula [6] as the adjusted contribution asset, $CA \cdot (1 + \alpha)$ (plus an initial capital fund, which we leave out for simplicity) equals the net liabilities of the system, i.e., the present value of net benefits, $PVBL - PVCL \cdot (1 + \alpha)$.

In the $GAC$ concept, the intertemporal balance sheet is calculated over an infinite time horizon [20,21]. Hence, in addition to contributions and benefits of living cohorts, the $GAC$ approach also takes into account contribution payments and benefits receipts of future cohorts. In the initial balance sheet of the $GAC$ concept, this is reflected by two additional items: the present values of contribution payments ($PVCF$) and the benefit receipts ($PVBF$) of future cohorts born after the base year $BY$. Similar to Equations (1) and (2), these two items are given by:

$$PVCF = \sum_{t=BY+1}^{\infty} N_{0,t} \sum_{u=0}^{100} ndf^{u+t-BY} \cdot l_{0,t,u} \cdot c_u \tag{9}$$

$$PVBF = \sum_{t=BY+1}^{\infty} N_{0,t} \sum_{u=0}^{100} ndf^{u+t-BY} \cdot l_{0,t,u} \cdot (1+cp)^{u+t-BY} \cdot b_u \tag{10}$$

The deficit of the intertemporal balance sheet in the case of the $GAC$ concept, which is referred to as the intertemporal public liabilities, $IPL$, is therefore given by:

$$IPL = PVBL + PVBF - PVCL - PVCF \tag{11}$$

Equation (12) shows the adjustment of contribution payments necessary to balance intertemporal assets and liabilities, $\theta$, which serves as our indicator for the $GAC$ concept:

$$\theta = \frac{IPL}{PVCL + PVCF} \tag{12}$$

Our third concept, $GAIB$ is derived from a generational accounting approach by Felder [22], which is often applied in studies concerned with the intergenerational distribution effects of reforms. Thereby these studies consider both the reform induced effects on future benefits as well as on future contribution payments (see e.g., [23–25]).

Thus, in addition to the contribution payments by living cohorts, the $GAIB$ balance sheet includes as a further asset the present value of "adjusted" contribution payments by living cohorts, $PVadjCL$. As pay-as-you-go schemes are (in general) not allowed to run deficits in the long run, the contribution payments in the $GAIB$ approach are computed by taking into account that contributions are adjusted in all future years to balance SHI revenues and expenditures on an annual basis. Hence, by taking future changes of contribution payments into account, the $GAIB$ approach attributes part of the underfunding in the $SM$ concept or the intertemporal public liabilities in the $GAC$ concept

to living cohorts. The additional payments by living cohorts, which result from future adjustments of contribution payments, are given by:

$$PVadjCL = \sum_{x=0}^{100} N_{x,BY} \sum_{u=x}^{100} ndf^{u-x} \cdot l_{x,BY,u} \cdot c_u \cdot adjc_{BY+u-x} \tag{13}$$

where $adjc_{BY+u-x}$ denotes the amount by which base year contributions have to be adjusted to balance the SHI budget in year $u$. Note that the calculation of $adjc_{BY+u-x}$ depends not only on the base year population size, but also on the size of future cohorts, which has an immediate impact on the contribution base in future years.

Similar to the other approaches, the intertemporal deficit under the *GAIB* concept, which we label implicit burden, *IB*, is given by:

$$IB = PVBL - PVCL - PVadjCL \tag{14}$$

It can be interpreted as the burden current generations are leaving to future generations via the pay-as-you-go financing in absolute numbers under the (pay-as-you-go inherent) assumption that they will be, along with future generations, faced with future contribution adjustments. In order to compare this result with the other measuring concepts, we calculate the additional increase of contributions, $\beta$, necessary to close the balance sheet:

$$\beta = \frac{IB}{PVCL + PVadjCL} \tag{15}$$

### 2.2. Methodological Similarities and Differences

Our measurement concepts can be grouped under the main approaches for measuring liabilities and assets in pension schemes [26–29]. Within this context, the *GAIB* concept resembles the so-called closed group with future accruals approach, as it concerns future payment flows only for cohorts alive in the base year. The *GAC* concept, however, follows an open group approach, as it measures future liabilities of both living and future cohorts.

The closed group without future accruals approach constitutes the starting point for applications of the work of Settergren and Mikula [6] because only pension entitlements of pensioners and contributors accrued to date are considered. Examples are the automatic balance mechanism of the Swedish pension system [7] as well as applications of the Settergren and Mikula [6] approach to the case of the defined benefit pension schemes in Spain, Germany, and Switzerland [8–11]. In contrast to the cited examples, which assess the liabilities of a pension scheme using the so-called "retrospective method", our application of the *SM* concept assesses the liabilities of the German SHI by computing the present value of future benefits and contribution payments, which Vidal-Meliá and Boado-Penas [30] call the "the prospective method". However, as pointed out by Ventura and Vidal-Meliá [31] and others [10,11,32], by taking the contribution asset into account, the Settergren and Mikula [6] approach is basically equivalent to an open group approach.

The three concepts differ not only regarding the items included in the intertemporal balance sheet but also regarding the underlying assumptions concerning the demographic transition, as well as future growth and interest rates. Settergren and Mikula [6] derive their condition of intergenerational fairness for the case of a stable population. Furthermore, they point out that the adequate discount factor in the case of a pay-as-you-go system is its internal rate of return $(1+g) \cdot (1+\eta)$, where $\eta$ denotes the annual rate of fertility-driven population growth. This reasoning goes back to Aaron [33]. We refer to these assumptions in our analysis in Section 3 with calculations based on a stationary population and a net discount factor $ndf$ equal to 1.

In contrast to the *SM* concept, the assumption of a real interest rate $r > g + \eta$ is central to concepts that measure fiscal sustainability over an infinite horizon, as in the *GAC* concept. In the context of the neoclassical growth models by Solow [34] and Diamond [35], this assumption is usually justified in that it implies that the economy is on a balanced

growth path that is *dynamically efficient*. While this assumption may be justified in the long run, one has to bear in mind that from a methodological point of view, the assumption $r > g + \eta$ is also a necessary condition for the application of generational accounting over an infinite horizon.

Because of the differing assumptions on the net discount factor, the standard *SM* approach cannot be directly compared with generational accounting approaches. Therefore, we use a modified approach to calculate the contribution asset, or rather the turnover duration, to allow for more general assumptions on the size of the net discount factor, i.e., $ndf < 1$. In detail, the calculation of the turnover duration components $A^B$ and $A^C$ have to be modified as follows:

$$A^B = \frac{\sum_{x=0}^{100} \frac{1+\xi}{\xi} \cdot \left[(1+\xi)^x - 1\right] \cdot N_x \cdot (1+cp)^x \cdot b_x}{\sum_{x=0}^{100} N_x \cdot (1+cp)^x \cdot b_x} \tag{16}$$

$$A^C = \frac{\sum_{x=0}^{100} \frac{1+\xi}{\xi} \cdot \left[(1+\xi)^x - 1\right] \cdot N_x \cdot c_x}{\sum_{x=0}^{100} N_x \cdot c_x} \tag{17}$$

where $\xi = ndf \cdot (1+\eta) = ((1+g)/(1+r)) \cdot (1+\eta)$. Hence, in contrast to Equations (5) and (6), the average value-weighted age of living beneficiaries in the base year *BY*, $A^B$, and contributors, $A^C$, includes the yearly population growth rate $\eta$ as well as the differential between the growth rate of contributions and benefits, *cp*, which we use in the analysis of Section 3. Please note that $\xi = 1$ if $(1+r) = (1+g) \cdot (1+\eta)$. As $\lim_{\xi \to 1} \left( \frac{1+\xi}{\xi} \left[(1+\xi)^x - 1\right] \right) = x$, it becomes apparent that Equations (5) and (6) are special cases of Equations (16) and (17).

### 2.3. Data and Assumptions

Our analysis includes three different population projections, calibrated age- and gender-specific profiles for contribution payments, and benefit receipts of SHI, as well as assumptions on the growth rate of productivity and the discount rate. As our base year, we choose 2018.

The input data of our three population scenarios, *stationary population*, *base year structure*, and *G2-L2-W0*, stem from Destatis [36] and the Max Planck Institute for Demographic Research [37]. Figure 1 shows the three population projections for the years 2018, 2068, and 2118 (scaled at the maximum value of age- and gender-specific cohorts in the corresponding population projection).

The scenario *stationary population*, with constant absolute births and constant life expectancy on basis of the life (Table 2016/18), serves as a benchmark for measuring the amount of "natural intergenerational distribution" in SHI. The second scenario, *base year structure*, starts from the base year population structure of Germany in 2018. For the projection, we assume constant survival rates as well as a constant absolute replacement at age 0 for men and women. Starting from the current age distribution, the population will transition into a stationary population within the next 101 years due to the assumptions of a maximum lifespan of 100 years as well as constant replacement and survival rates. Compared with the stationary population, this second scenario serves to measure the impact of Germany's current demographic composition on intergenerational fairness.

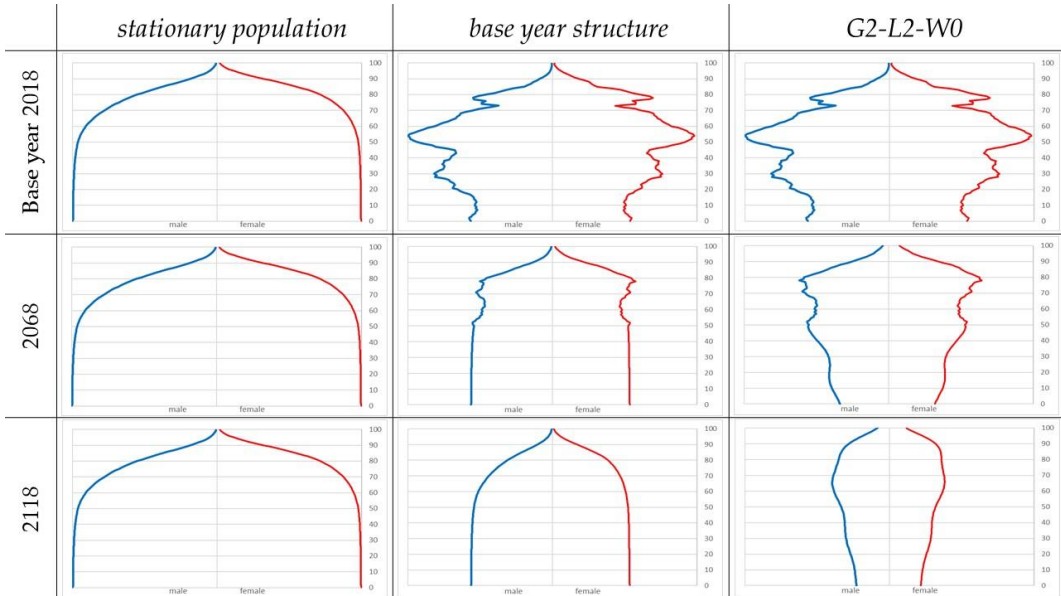

**Figure 1.** Population projections in the years 2018, 2068, and 2118. Note: male on the left side (blue), female on the right side (red).

Our third scenario, called *G2-L2-W0*, is adapted from the 14th population projection of Destatis [36]. In line with the fertility assumption G2 and mortality assumption L2 of the official forecast, we assume a constant total fertility rate of 1.55 children per woman of childbearing age and an increase in the life expectancy at birth from 83.3 (78.5) in the year 2018 to 88.1 (84.4) in the year 2060 for women (men). To focus on the impact of low fertility and rising life-expectancy on intergenerational fairness, we neglect the impact of future migration, an assumption that we term "W0". Compared with the scenario *"base year structure"*, this third population forecast will serve to measure the impact of longevity on intergenerational fairness in SHI.

For the calculation of average benefit receipts, $b_u$, we use calibrated age- and gender-specific micro profiles for the year 2018, which stem from the morbidity-oriented risk structure compensation scheme of German SHI [38]. As a result of the applied calibration, the total sum of benefits for the SHI-insured population 2018 in our model matches the SHI expenditures of EUR 239.4 billion in the year 2018 [39].

In order to compute the initial average contribution payments, $c_u$, of 2018, we use age- and gender-specific data on SHI contribution payments from the German Income and Consumption Survey (Einkommens- und Verbrauchsstichprobe) of Destatis [40]. Note that for simplicity, our analysis is based on the assumption that the SHI scheme exhibits a balanced budget in the base year. To this end we calibrated the age-profiles, such that the total amount of contribution payments in our model is equal to the total amount of benefits in the base year 2018. Figure A1 in the Appendix A shows the resulting age- and sex-specific contribution payments and benefits of the German SHI ($c_u$ and $b_u$).

As mentioned in the previous section, we refer to the methodological framework of Settergren and Mikula [6] with a net discount factor $ndf$ equal to one. In line with the assumptions used in generational accounting approaches for Germany and other countries (e.g., [41,42]) as well as in the sustainability report of the European Commission [43] we set a productivity growth rate $g = 1.5\%$ and a real interest rate $r = 3\%$ p.a., which results in a $ndf = (1+g)/(1+r) = 1.015/1.03 = 0.985$. In order to check the robustness of our results, we conduct a sensitivity analysis with differing assumptions for the net discount factor.

For our scenario of an ongoing cost pressure due to the medical technical progress, we assume a differential between the growth rates for contributions and benefits of 0.5 percentage points, which is in line with the range of empirical estimates. For Ger-

many, empirical studies (e.g., [13,14]) estimate that the annual growth rates of health expenditures per capita outgrows the growth rate of GDP per capita by about one percentage point. As our projection period considers current generations over the next 101 years, which seems to be very long, for the following calculations we choose a growth of benefits that is 0.5 percentage points higher than contributions.

## 3. Results

### 3.1. The "Natural" Amount of Intergenerational Redistribution

In the case of a stationary population and a net discount factor of 0.985, the results of our analysis are presented in Table 2. All absolute values are normalized on the present value of benefits for living generations, $PVBL$, (times 100).

**Table 2.** Intertemporal balance sheets—Stationary population, ndf = 0.985, 2018. Note: All nominal values are normalized so that the present value of future benefit receipts by living generations (PVBL) is equal to 100. For the computations of present values, a net discount factor (ndf) of 0.985 is applied.

| Item | SM Assets | SM Liabilities | GAC Assets | GAC Liabilities | GAIB Assets | GAIB liabilities |
|---|---|---|---|---|---|---|
| PVCL | 96 | | 96 | | 96 | |
| PVBL | | 100 | | 100 | | 100 |
| CA | 4 | | | | | |
| PVCF | | | 94 | | | |
| PVBF | | | | 90 | | |
| PVadjCL | | | | | 0 | |
| UF/IPL/IB | 0 | | 0 | | 4 | |
| $\alpha/\theta/\beta$ | | 0.0% | | 0.0% | | 4.6% |

In the case of a stationary population, both the *SM* approach and *GAC* lead to the same result. With a (normalized) amount of intergenerational redistribution of approximately four percent of living generations, future benefit receipts are paid for by future generations. In the case of the *SM* approach, the amount of intergenerational redistribution is directly reflected in the contribution asset *CA*, while in the case of the *GAC* approach, it is visible as the difference between the present value of future generations' contribution payments and benefit receipts.

Moreover, under the assumption of a stationary population and identical growth rates for benefits and contributions, both approaches indicate that the system is stable, i.e., intertemporal assets and liabilities are in balance, and there is no need to increase contribution payments.

In contrast, in the *GAIB* approach, the balance sheet shows a deficit amounting to 4. The indicator $\beta$ would therefore demand an increase of contributions by 4.6 percent to balance intertemporal assets and liabilities. This result casts some doubt on the applicability of the *GAIB* approach as a measure of intergenerational redistribution implied by pay-as-you-go schemes. The reason is that the *GAIB* approach does not take into account that, by definition, a pay-as-you-go-system implies some amount of intergenerational redistribution. On the contrary, the *GAIB* approach is more in line with a market-based health insurance scheme, where the contribution payments of each generation or cohort are only utilized to pay for benefits received by its members. Under these circumstances, however, each generation or cohort will accumulate an asset that when young will be sufficient to pay for higher benefit receipts when old. In terms of the balance sheet this asset would be sufficient to counterbalance the deficit shown in Table 2.

Finally, as shown in Table A1 in Appendix B, these implications are qualitatively the same in the case of a net discount factor of 1. The major difference is that the *GAIB* approach indicates an even larger amount of intergenerational redistribution compared with the case of a net discount factor of 0.985. In line with a market-based approach, however, the larger amount of *IB* simply reflects the fact that with a lower interest rate a larger amount of contributions must be saved when young to pay for benefits when old.

### 3.2. The Impact of Demographic Change on Intergenerational Redistribution

The intertemporal balance sheets resulting from the three different population projections can be found in Table A1 in Appendix B. Figure 2 shows the indicators for the three measuring concepts in the case of a net discount factor of 0.985. As outlined in Section 2, the scenario *base year structure* thereby shows the impact on intergenerational redistribution implied by today's age distribution of the population or by past changes in fertility, mortality, and migration rates, while scenario *G2-L2-W0* also takes the impact to the projected future increase in life expectancy into account.

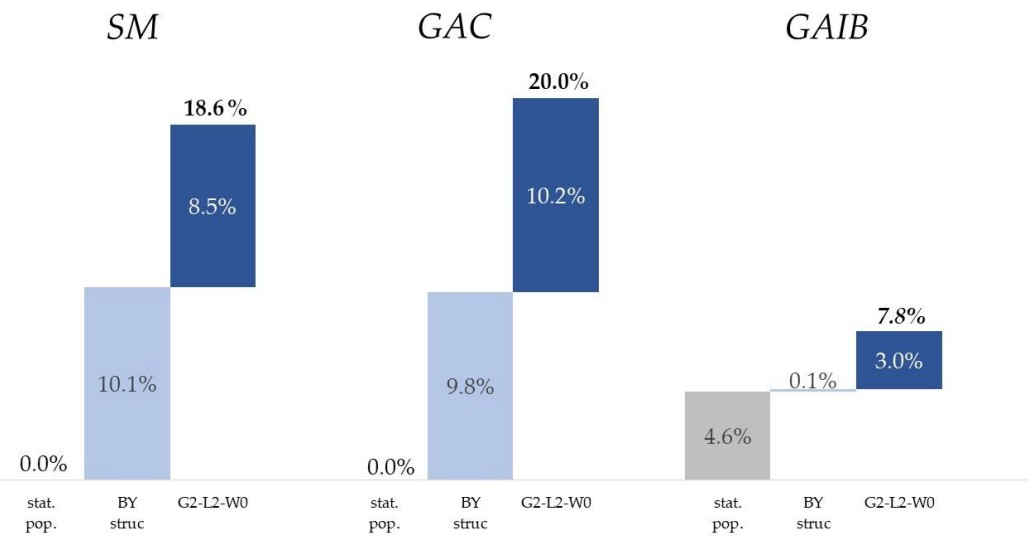

**Figure 2.** Indicators for different population projections, ndf = 0.985.

Compared with the results for the stationary population, both the *SM* and the *GAC* approach imply a deficit in the intertemporal balance sheet in approximately the same order of magnitude. In the scenario *base year structure,* the *SM* approach demands an increase in today's contribution rate of 10.1 percent (α), while the *GAC* approach demands a slightly smaller increase of 9.8 percent (θ) to balance the intertemporal balance sheet. Additionally, taking into account the future increase in life expectancy shows an even larger amount of intergenerational redistribution. In scenario *G2-L2-W0,* the *SM* approach demands an increase in today's contribution rate of 18.6 percent (α), and the *GAC* approach again needs a slightly higher increase of 20.0 percent (θ) to balance the intertemporal balance sheet.

In contrast to *SM* and *GAC,* the *GAIB* approach implies considerably smaller amounts of intergenerational redistribution. For the scenario *base year structure,* the intertemporal deficit is in the same order of magnitude as in the case of a stationary population. Taking into account the increase in life expectancy, the *GAIB* approach also implies an increase in intergenerational redistribution. However, the increase is considerably smaller than in the case of the *SM* and *GAC* approaches.

These results first indicate that under identical assumptions both the *SM* and *GAC* approach lead to similar results and conclusions regarding the amount of intergenerational redistribution. Second, all three approaches show that longevity is a major driver of intergenerational redistribution in SHI. Even in the *GAIB* approach, where the projected increase in contribution rates is already taken into account, our results show that the burden of longevity is partly shifted to future cohorts of contribution payers. The reason is that a rising life expectancy also means additional years for living generations, in which their average benefits exceed their average contribution payments.

### 3.3. The Influence of an Ongoing Cost Pressure

A key characteristic of our results presented so far is the assumption of a uniform growth rate for age-related contributions and benefits. However, in the last five decades

health expenditures in most industrial countries have grown at a significantly higher rate than GDP per capita [12]. Figure 3 shows the impact of an additional cost pressure on our indicators $\alpha$, $\theta$, and $\beta$, under the assumption of the population scenario *G2-L2-W0*.

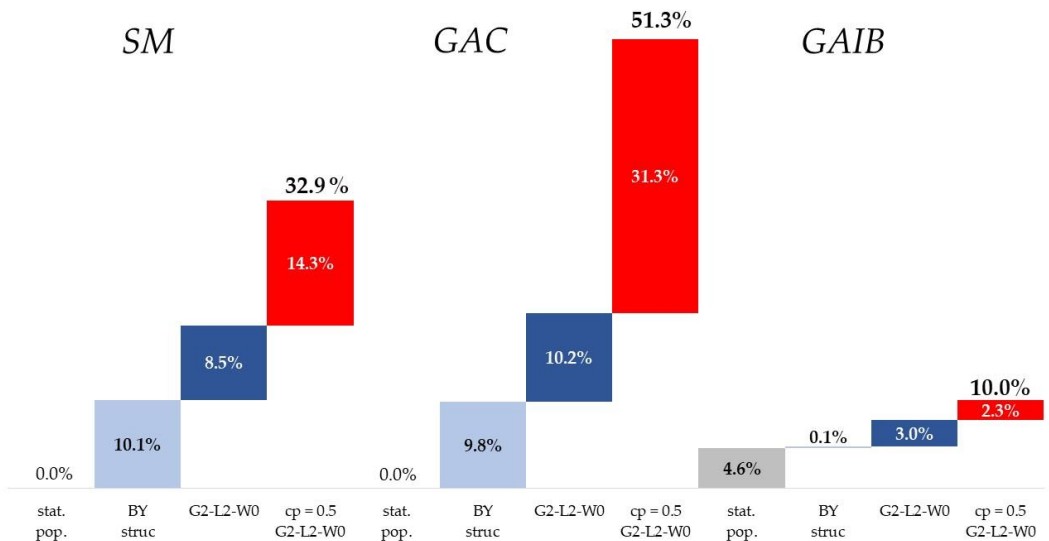

**Figure 3.** Indicators for different population projections and cost pressure, ndf = 0.985.

In the *SM* concept, $\alpha$ amounts to 32.9 percent, which is 14.3 percentage points higher than in the scenario without cost pressure. The indicator $\theta$ of the *GAC* concept amounts to 51.3 percent, which is 31.3 percentage points higher than in the scenario without cost pressure. Thus, in the cost pressure scenario, $\theta$ now clearly exceeds $\alpha$. The reason for this is that in the *GAC* concept the rise in liabilities due to medical progress (i.e., the higher growth rate of benefits in the cost pressure scenario) leads both to a rise of benefits received by present living (*PVBL*) and future generations (*PVBF*).

In contrast to the *SM* and *GAC* concepts, cost pressure induces only a comparatively small increase in the *GAIB* concept. In this concept $\beta$ rises to 10.0 percent, which is 2.3 percentage points higher compared with the scenario without cost pressure. The reason for this small increase is again the asset *PVadjCL*. At least partially, living generations have to pay for the additional benefits they will receive due to medical progress because they will also be faced with (under the cost pressure scenario) higher contribution rates during their remaining lifetime.

Additionally, the other population projections show approximately the same impact of cost pressure on the results. This can be seen in the Appendix B by comparing the results in Table A1 (intertemporal balance sheets with cost pressure) with those in Table A2.

### 3.4. Sensitivity Analysis

As mentioned in Section 2.2, one main difference between the "world of generational accounting" and the Settergren and Mikula [6] approach concerns the assumptions on growth and interest. Thus, we carry out a sensitivity analysis on this assumption and calculate the indicators for net discount factors in the range of 0.975 and 0.995. The results of this sensitivity analysis can be found in Appendix C in Figure A2 (scenario without cost pressure) and Figure A3 (scenario with cost pressure).

As one can see from Figure A2, the sensitivity analysis confirms our results so far: Under the assumption of a stationary population, the indicators of the *SM* concept ($\alpha$) as well as the *GAC* concept ($\theta$) show no adjustment, independent of the chosen net discount factor. Moreover, the values of the indicators $\alpha$ and $\theta$ are in a similar range under the population projections *base year structure* and *G2-L2-W0*, and the net discount factor than lower than 0.985 as well. However, for a net discount factor above 0.985, $\theta$ shows a sharp increase compared with $\alpha$ under the population projection *G2-L2-W0*. Thus, it can be stated

that only for large differences between growth and interest rates (i.e., a comparable small net discount factor) the *GAC concept* and the *SM concept* lead to similar results.

The indicator $\beta$ of the GAIB concept shows a linear increase for all three population scenarios when the net discount factor increases. This again is due to the implicit market-based interpretation that with a lower interest rate (i.e., a higher net discount factor) a larger amount of contributions must be saved when young to pay for benefits when old.

For the cost pressure scenario (Figure A3), these results are qualitatively the same, highlighting the sensitivity of the indicator $\theta$ on the chosen growth and interest rates.

## 4. Discussion—A Final Statement on the Three Measurement Concepts

Our results are summarized in Table 3. Even though the *GAC* and the *SM* concept are derived from different theoretical backgrounds, their results are very close in all scenarios without cost pressure when there is a significant difference on the chosen parameters for interest and growth.

**Table 3.** Comparison of the measurement concepts.

| | **SM Concept** | **GAC Concept** | **GAIB Concept** |
|---|---|---|---|
| Basis concept | Settergren/Mikula [6] | Classical generational accounting approaches | Generational accounting with intergenerational distribution effects |
| Theoretical background *Assumptions* Original objective of measuring | Aaron [33] $(1+r)=(1+g)\cdot(1+\eta)$ Balance rule | Solow [34] and Diamond [35] $r > g + \eta$ Fiscal sustainability | Solow [34] and Diamond [35] $r > g + \eta$ Intergenerational distribution effects |
| Considered assets in the intertemporal budget sheet | Contribution asset | Present value of future generations contributions and benefits from pay-as-you go | Necessary increase (or decrease) of contributions to reach annually balanced budgets |
| **Indicator** | $\alpha$ | $\theta$ | $\beta$ |
| Results indicator *stationary population* | No adjustment | No adjustment | Increase (net benefits of present generations have to be financed by future generations) |
| *Results indicator population base year structure* | *Necessary adjustment (the current German population structure will ceteris paribus induce a higher benefit contribution ratio)* | *Necessary adjustment (the current German population structure will ceteris paribus induce a higher benefit contribution ratio)* | *A priori uncertain (higher benefits for living generations are compensated through their higher contribution payments)* |
| Results indicator increase of life expectancy (*G2L2W0* vs. *base year structure*) | Necessary adjustment as the current German population structure will ceteris paribus induce a higher benefit contribution ratio | Necessary adjustment, as the current German population structure will ceteris paribus induce a higher benefit contribution ratio | Necessary comparatively low adjustment (higher benefits for living generations are partly compensated through their higher contribution payments) |
| *Results indicator cost pressure due to medical progress* | *Necessary adjustment, comparable with the demographic components* | *Necessary adjustment that is higher than the effect of demographic components because future generations also benefit from the medical progress* | *Necessary comparatively low adjustment (higher benefits for living generations are partly compensated through their higher contribution payments)* |
| Sensitivity analysis (variations of growth and interest rates) | Comparatively robust results | Comparatively robust results when there is a significant difference between $g$ and $r$; Comparatively less robust results when there is a small difference between $g$ and $r$ | Linear increase of $\beta$ with the net discount factor due to the market-based assumption that a larger amount of contributions must be saved when young to pay for benefits when old |

In the case of the *stationary population*, $\theta$ and $\alpha$ equal zero. The reason is that both methods consider the "natural" amount of intergenerational distribution. In the context of a defined benefit pension system, this result can also be found in Vidal-Meliá and Boado-

Penas [30], who compare the contribution asset with the hidden asset that results within a market-based intertemporal measurement concept. As they point out, the hidden asset can also be interpreted as a pay-as-you go asset, which is also known as a hidden or implicit tax in the literature on pension systems.

In the case of an aging population, $\alpha$ and $\theta$ show a similar degree of intergenerational fairness. However, in the cost pressure scenario, $\theta$ exceeds $\alpha$ by far. Thus, while the GAC approach seems appropriate as a measure of the fiscal sustainability of public or social insurance finances, it seems unsuitable as a statement of intergenerational fairness.

For the *GAIB* concept, it can be stated that it is quite intuitive at first sight. At a second glance, however, this concept shows shortcomings as a (global) statement of intergenerational fairness. Hence, this concept might be appropriate as a measure of the distributional effects of policy reforms, but it seems unsuitable as a global indicator for intergenerational fairness.

As we pointed out, the *SM* concept seems to be adequate for measuring the degree of intergenerational fairness in SHI. Furthermore, in the light of the ongoing debate regarding low interest rates [44,45], the *SM* concept offers an advantage over the other measurement concepts, as the assumption $r = g$ could presumably be more comprehensible to a broader audience than the assumption of $r$ far exceeding $g$.

Of course, our results are also subject to limitations. In particular, the extrapolation of age- and gender-specific payment flows according to the status quo principle (i.e., assuming constant age-related contribution payments and benefit receipts, both growing at a uniform rate) can be criticized. On the SHI revenue side, it can be assumed that people who retire later will pay more contributions in the future. Similarly, a higher proportion of women can be expected to be employed in the future and will thus pay more contributions [46]. However, the impact of other assumptions about the future development of contributions on our results should be small, as there will be fewer generations of taxpayers in the near future due to the demographic transition [14].

With regard to the expenditure side of SHI in particular, the effect of an increase in life expectancy on the age-related health expenditures seems to be unclear and has been part of a controversial debate in the health economic literature over the last 20 years [47]. Zweifel et al. [48–50] call the observed correlation between rising age and rising health care expenditure a "red herring" because the majority of this correlation is due to high health care spending at the end of life. Extending life expectancy would therefore simply lead to high health care spending at a later point in time. As a result, the application of the status quo principle would predict health care expenditures that are too high. However, it is questionable whether this conclusion can really be drawn. As Breyer and Lorenz [47] argue, there are many reasons to believe that the aging of society will lead to rising health care expenditures in the future. In our study, this discussion is reflected in the range of our results for the population scenarios *G2L2W0* and *base year structure*.

Another limitation, of course, concerns the assumed growth differential in our cost-pressure scenario. In the work of Newhouse [51], the sharp increase in health care spending was mostly attributed to technological change in medical care. Following this work, many scholars have developed models to explain how technological change in health care occurs (for an overview, see [16]). More recent approaches show that the extent of medical progress varies widely for different diseases [52,53]. However, their future development also depends, in turn, on demographic factors [54]. Overall, there are many different factors interacting here, and more research is needed to predict future trends in health care spending more accurately.

In addition to technological advances, however, the access to and the generosity of health insurance systems has been another important factor contributing to the growth in health care spending in the past [55]. In the end, the extent of future health care spending depends on the generosity of SHI, which in turn depends on society's willingness to finance an expensive social insurance system.

## 5. Conclusions

In this paper we considered different (potential) methods for measuring the degree of intergenerational (un-)fairness in SHI. To the best of our knowledge, this is the first application of the Settergren and Mikula [6] approach to the measurement of intergenerational fairness in the German SHI. Despite different methodological backgrounds, our results indicate that the *SM* concept based on Settergren and Mikula [6] leads to similar implications concerning the amount of intergenerational redistribution as classical measures of fiscal sustainability.

With regard to intergenerational fairness, our results show that currently SHI-insured persons will leave a burden on future generations. To close the intertemporal balance sheet of the SHI requires an increase of contribution payments by 30 to 40 percent. This total effect can be separated into an effect due to the current base year structure of the German population (10 p.p.), an effect due to the increase of life expectancy (10 p.p.), and an effect due to medical technical progress (about 10 to 20 p.p.). The first and the latter effect seem to be unquestioned. However, the second effect is much more uncertain, as the impact of longevity on future health care costs is part of a controversial debate in the health economics literature.

The policy implications of our results foremostly concern the future design of the German health system. Following the example of many pension systems worldwide, we propose installing a regular annual calculation of intertemporal balance sheets and a balance mechanism for the German SHI as well as for the German long-term-care insurance scheme, which is even more affected by the challenge of the demographic transition. Such a financial framework would force policymakers to take the long-term consequences of current health policy and health care reform plans into account.

The transfer of our analysis to other countries primarily concerns those with a statutory health care system in the Bismarckian tradition and a rapidly aging population (e.g., Japan, France, and Austria). In principle, however, an intertemporal budgeting system for public health expenditures could also be established in Scandinavian countries or the United Kingdom with a tax-funded national health service in the Beveridge tradition (using a "tax asset" instead of a contribution asset). Indeed, the traditional settings of health systems have in many ways dissolved in the past decades [56], and almost all industrialized nations are affected by the challenges of population aging and technological change in health care.

Our findings also have policy implications for low/middle income states, such as the BRICS countries in particular. Although public health care spending has risen only moderately in these countries in the past [57], this is likely to change in the future. One crucial condition for future prosperity is universal health coverage, which in turn can only be achieved through an increase in public health care spending [58,59]. Furthermore, countries such as China are also facing the challenge of an aging society due to low birth rates in the 1980s [60,61]. Therefore, intergenerational fairness could be one important aspect in the discussion on the optimal design of public health care schemes in these countries [62].

**Author Contributions:** Conceptualization, S.M.; methodology S.M. and S.F.; calculation S.F.; writing, S.M. and S.F. Both authors have read and agreed to the published version of the manuscript.

**Funding:** This research received no external funding.

**Institutional Review Board Statement:** Not applicable.

**Informed Consent Statement:** Not applicable.

**Data Availability Statement:** Population data are available at Destatis, Germany's official statistical office, and mortality.org. The data for the calculation of age- and gender-specific micro profiles are available at the Federal Social Insurance Authority of Germany. Statistics about the SHI Finances are available at the German Federal Ministry of Health. The calculations used in this paper are available upon request.

**Acknowledgments:** We are grateful to Jana Wolf for helpful comments and criticism.

**Conflicts of Interest:** The authors declare no conflict of interest.

## Appendix A

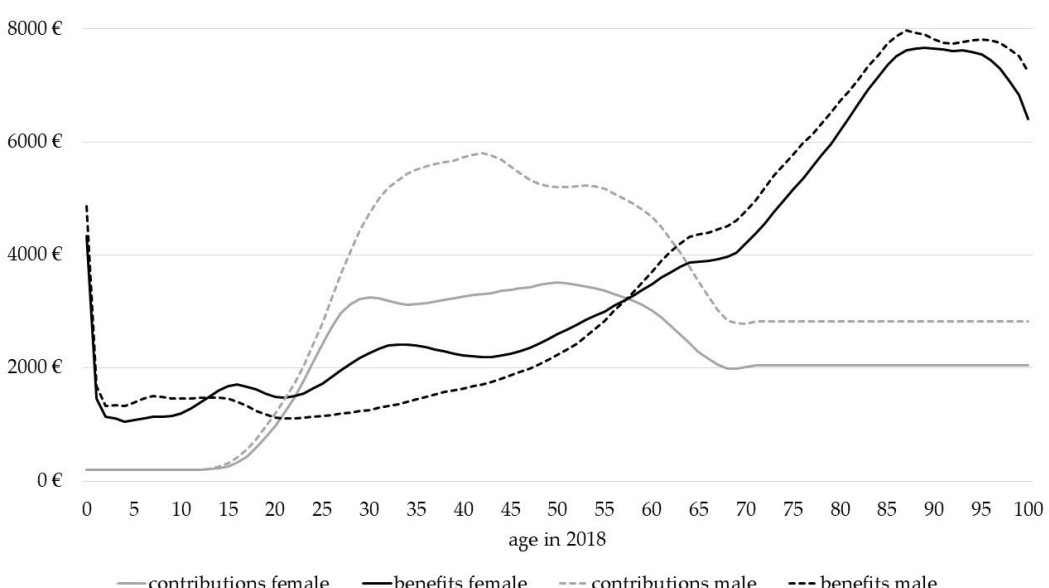

**Figure A1.** Age- and sex-specific benefits and contributions in SHI 2018.

## Appendix B

**Table A1.** Intertemporal balance sheets without cost pressure.

| | Stationary population, ndf = 1 | | | | | | Stationary population, ndf = 0.985 | | | | | |
|---|---|---|---|---|---|---|---|---|---|---|---|---|
| | **SM** | | **GAC** [1] | | **GAIB** | | **SM** | | **GAC** | | **GAIB** | |
| **Item** | A | L | A | L | A | L | A | L | A | L | A | L |
| PVCL | 89 | | 89 | | 89 | | 96 | | 96 | | 96 | |
| PVBL | | 100 | | 100 | | 100 | | 100 | | 100 | | 100 |
| CA | 11 | | | | | | 4 | | | | | |
| PVCF | | | n.a. | | | | | | 94 | | | |
| PVBF | | | | n.a. | | | | | | 90 | | |
| *PVadjCL* | | | | | 0 | | | | | | 0 | |
| *UF/IPL/IB* | 0 | | n.a. | | 11 | | 0 | | 0 | | 4 | |
| *α/θ/β* | 0.0% | | n.a. | | 12.1% | | 0.0% | | 0.0% | | 4.6% | |
| | **Base year structure, ndf = 1** | | | | | | **Base year structure, ndf = 0.985** | | | | | |
| | **SM** | | **GAC** | | **GAIB** | | **SM** | | **GAC** | | **GAIB** | |
| **Item** | A | L | A | L | A | L | A | L | A | L | A | L |
| PVCL | 81 | | 81 | | 81 | | 86 | | 86 | | 86 | |
| PVBL | | 100 | | 100 | | 100 | | 100 | | 100 | | 100 |
| CA | 10 | | | | | | 5 | | | | | |
| PVCF | | | n.a. | | | | | | 68 | | | |
| PVBF | | | | n.a. | | | | | | 69 | | |
| PVadjCL | | | | | 9 | | | | | | 9 | |
| UF/IPL/IB | 9 | | n.a. | | 10 | | 9 | | 15 | | 5 | |
| *α/θ/β* | 10.1% | | n.a. | | 11.0% | | 10.1% | | 9.8% | | 4.7% | |

**Table A1.** *Cont.*

| Item | G2_L2_W0, ndf = 1 SM A | L | GAC A | L | GAIB A | L | G2_L2_W0, ndf = 0.985 SM A | L | GAC [2] A | L | GAIB A | L |
|---|---|---|---|---|---|---|---|---|---|---|---|---|
| PVCL | 74 | | 74 | | 74 | | 81 | | 81 | | 81 | |
| PVBL | | 100 | | 100 | | 100 | | 100 | | 100 | | 100 |
| CA | 7 | | | | | | 3 | | | | | |
| PVCF | | | n.a. | | | | | | 40 | | | |
| PVBF | | | | n.a. | | | | | | 46 | | |
| PVadjCL | | | | | 12 | | | | | | 11 | |
| UF/IPL/IB | 19 | | n.a. | | 13 | | 16 | | 24 | | 7 | |
| $\alpha/\theta/\beta$ | 23.5% | | n.a. | | 15.5% | | 18.6% | | 20.0% | | 7.8% | |

[1] Note that the GAC approach is not feasible or applicable in the case of ndf = 1 because the present values of future generations' contributions and benefits are not defined due to the infinite horizon assumption. [2] In absolute terms the scenario *G2-L2-W0* with a $ndf = 0.985$ offers an *IPL* equal to EUR 2296 billion or 68.6 percent of German GDP in 2018. This is in line with other generational accounting studies focusing on the SHI scheme.

**Table A2.** Intertemporal balance sheets with cost pressure (*cp* = 0.5%).

| Item | Stationary population, ndf = 1 SM A | L | GAC[1] A | L | GAIB A | L | Stationary population, ndf = 0.985 SM A | L | GAC A | L | GAIB A | L |
|---|---|---|---|---|---|---|---|---|---|---|---|---|
| PVCL | 75 | | 75 | | 75 | | 83 | | 83 | | 83 | |
| PVBL | | 100 | | 100 | | 100 | | 100 | | 100 | | 100 |
| CA | 14 | | | | | | 8 | | | | | |
| PVCF | | | n.a. | | | | | | 82 | | | |
| PVBF | | | | n.a. | | | | | | 118 | | |
| PVadjCL | | | | | 12 | | | | | | 10 | |
| UF/IPL/IB | 11 | | n.a. | | 13 | | 9 | | 53 | | 6 | |
| $\alpha/\theta/\beta$ | 12.5% | | n.a. | | 15.4% | | 9.9% | | 31.7% | | 6.8% | |

| Item | Base year structure, ndf = 1 SM A | L | GAC A | L | GAIB A | L | Base year structure, ndf = 0.985 SM A | L | GAC A | L | GAIB A | L |
|---|---|---|---|---|---|---|---|---|---|---|---|---|
| PVCL | 69 | | 69 | | 69 | | 76 | | 76 | | 76 | |
| PVBL | | 100 | | 100 | | 100 | | 100 | | 100 | | 100 |
| CA | 12 | | | | | | 7 | | | | | |
| PVCF | | | n.a. | | | | | | 59 | | | |
| PVBF | | | | n.a. | | | | | | 91 | | |
| PVadjCL | | | | | 19 | | | | | | 18 | |
| UF/IPL/IB | 19 | | n.a. | | 12 | | 17 | | 56 | | 6 | |
| $\alpha/\theta/\beta$ | 23.5% | | n.a. | | 13.9% | | 20.8% | | 41.5% | | 6.6% | |

| Item | G2_L2_W0, ndf = 1 SM A | L | GAC A | L | GAIB A | L | G2_L2_W0, ndf = 0.985 SM A | L | GAC A | L | GAIB A | L |
|---|---|---|---|---|---|---|---|---|---|---|---|---|
| PVCL | 62 | | 62 | | 62 | | 70 | | 70 | | 70 | |
| PVBL | | 100 | | 100 | | 100 | | 100 | | 100 | | 100 |
| CA | 8 | | | | | | 5 | | | | | |
| PVCF | | | n.a. | | | | | | 35 | | | |
| PVBF | | | | n.a. | | | | | | 60 | | |
| PVadjCL | | | | | 22 | | | | | | 20 | |
| UF/IPL/IB | 30 | | n.a. | | 16 | | 25 | | 54 | | 9 | |
| $\alpha/\theta/\beta$ | 42.5% | | n.a. | | 19.0% | | 32.9% | | 51.3% | | 10.0% | |

Note that the GAC approach is not feasible or applicable in the case of ndf = 1 because the present values of future generations' contributions and benefits are not defined due to the infinite horizon assumption.

**Appendix C**

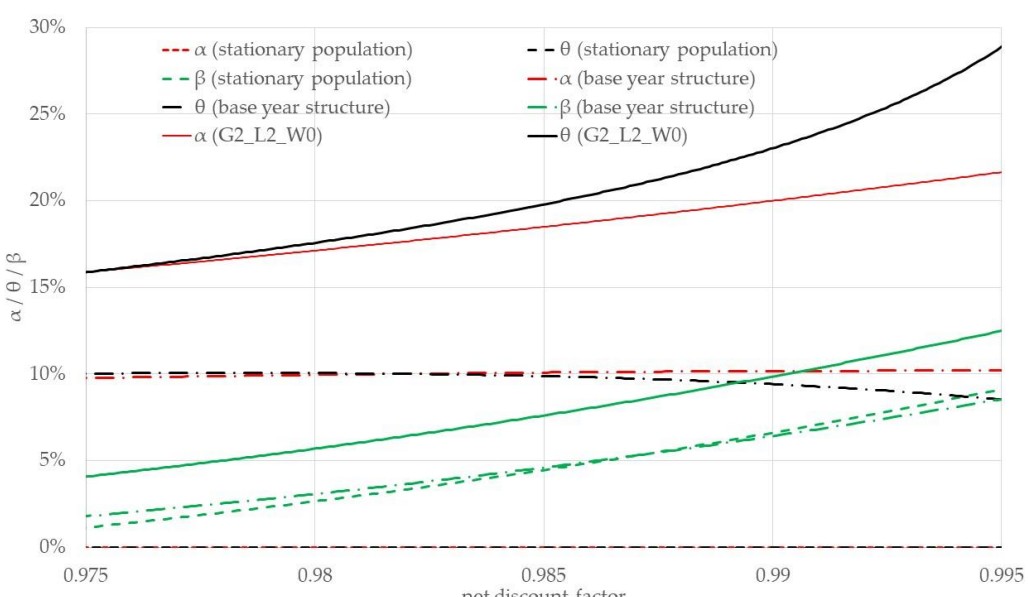

**Figure A2.** Sensitivity analysis, scenario without cost pressure.

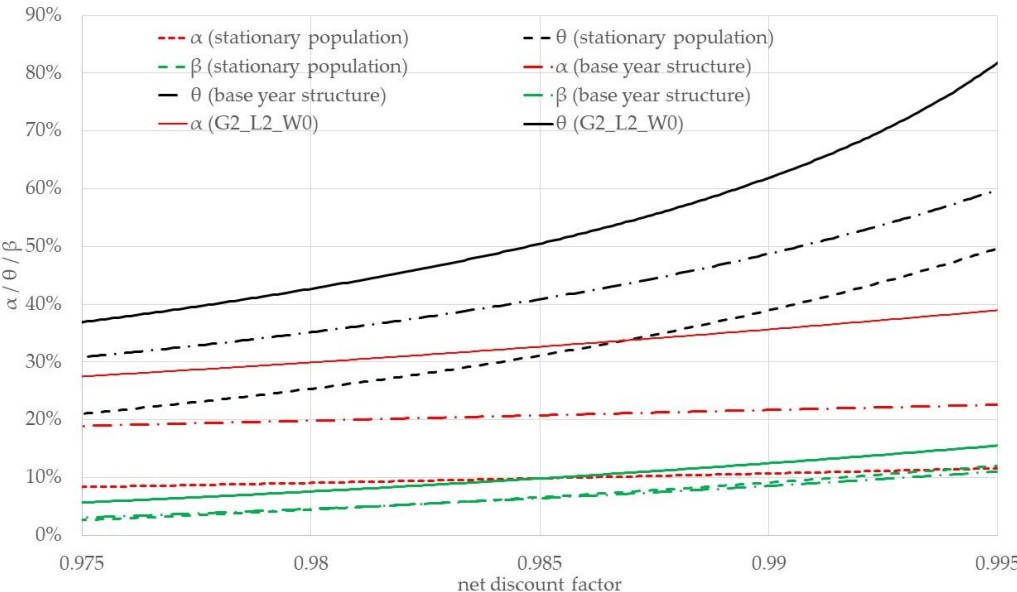

**Figure A3.** Sensitivity analysis, scenario with cost pressure (*cp* = 0.5%).

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
