# Peer review of "Indicators for Measuring Intergenerational Fairness of Social Security Systems—The Case of the German Social Health Insurance"

_sustainability, doi:10.3390/su13105743_

Round 1
Reviewer 1 Report
Dear Authors,
It was a pleasure to review your contribution entitled:
"Indicators for Measuring Intergenerational Fairness of Social Security Systems – The case of the German Social Health Insurance "
It may fill a certain knowledge gap in the existing health economic literature.
Yet it has one bottleneck weakness.
Body of evidence cited is too homogeneous leaning heavily towards Swiss, German and OECD academic sources.
Given the contemporary world momentum there is an increasing relevance of transnational comparability of German health insurance system with the ones of LMICs countries and leading Emerging Markets such as BRICS in particular, which remain the main pillar of real Global GDP growth and overall demand for medical goods and services.
CItations track record should thus be substantially diversified and expanded to encircle evidence outsourcing from these markets for the purpose of credible transnational comparisons.
Thus I would warmly recommend consideration for inclusion of at least several sources beneath:
https://www.sciencedirect.com/science/article/abs/pii/S0140673614600751
https://www.scielosp.org/article/bwho/2014.v92n6/429-435/en/
https://onlinelibrary.wiley.com/doi/abs/10.1002/hec.3406
https://www.frontiersin.org/articles/10.3389/fphar.2016.00021/full
https://academic.oup.com/heapol/article/31/6/717/1749704?login=true
https://resource-allocation.biomedcentral.com/articles/10.1186/s12962-020-00210-2
https://www.scielosp.org/pdf/bwho/2014.v92n6/394-395
https://www.ncbi.nlm.nih.gov/pmc/articles/PMC7585857/
www2.southeastern.edu/orgs/ijae/index_files/IJAE%20MARCH%202016%20KULKARNI%20-%20MARCH%2029%202016.pdf
https://www.mdpi.com/1660-4601/17/24/9404/htm
https://www.tandfonline.com/doi/full/10.3111/13696998.2015.1093493
https://papers.ssrn.com/sol3/papers.cfm?abstract_id=2725414
https://www.frontiersin.org/articles/10.3389/fpubh.2016.00002/full
https://www.tandfonline.com/doi/full/10.1080/13696998.2019.1600523
https://globalizationandhealth.biomedcentral.com/articles/10.1186/s12992-020-00590-3
Reviewer 2 Report
Please see the attachment.

Reviewer 3 Report
The “Indicators for Measuring Intergenerational Fairness of Social Security Systems – The case of the German Social Health Insurance” research paper is an extremely interesting paper. It deals with several aspects related to the fiscal sustainability (intergenerational fairness) of the German Social Health Insurance System, considering different methods to measure the degree of intergenerational fairness of this typical Bismarck healthcare system that faces certain future challenges.
I have closely read the entire paper and have a few comments for the authors.
-The sections are well-balanced, rather recent references are also used and these sections are well-grounded. I particularly appreciate the Methods and Data, and the Discussions subsections. Chapter 3 is the widest, including a sensitivity analysis.
-The theme of this research paper is interesting. I appreciate the comparison of the measurement concepts from Table 3, the discussions as well, but the Conclusions subchapter ends the paper abruptly. Will you please add some more policy implications and the international extrapolation of your findings. A short comparison to other EU healthcare systems that use the same predominant financing mechanisms would be interesting as well. These always increase the quality of a paper.
-How do you deal with the limits of your paper and which are your future research avenues? Please add these in the Conclusions as well.

Round 2
Reviewer 2 Report
This version is much better than the original. The revision of manuscript successfully addressed most of my comments.